# Comparative Study of Sample Carriers for the Identification of Volatile Compounds in Biological Fluids Using Raman Spectroscopy

**DOI:** 10.3390/molecules27103279

**Published:** 2022-05-20

**Authors:** Panagiota Papaspyridakou, Michail Lykouras, Christos Kontoyannis, Malvina Orkoula

**Affiliations:** 1Department of Pharmacy, University of Patras, GR-26504 Rio, Achaias, Greece; gioulipap@upatras.gr (P.P.); lykourasm@upnet.gr (M.L.); kontoyan@upatras.gr (C.K.); 2Institute of Chemical Engineering Sciences, Foundation of Research and Technology-Hellas (ICE-HT/FORTH), GR-26504 Platani, Achaias, Greece

**Keywords:** raman spectroscopy, carriers, sample holders, gold layer, cuvette, ethanol, urine, volatile compounds, biological fluids

## Abstract

Vibrational spectroscopic techniques and especially Raman spectroscopy are gaining ground in substituting the officially established chromatographic methods in the identification of ethanol and other volatile substances in body fluids, such as blood, urine, saliva, semen, and vaginal fluids. Although a couple of different carriers and substrates have been employed for the biochemical analysis of these samples, most of them are suffering from important weaknesses as far as the analysis of volatile compounds is concerned. For this reason, in this study three carriers are proposed, and the respective sample preparation methods are described for the determination of ethanol in human urine samples. More specifically, a droplet of the sample on a highly reflective carrier of gold layer, a commercially available cuvette with a mirror to enhance backscattered radiation sealed with a lid, and a home designed microscope slide with a cavity coated with gold layer and covered with transparent cling film have been evaluated. Among the three proposed carriers, the last one achieved a quick, simple, and inexpensive identification of ethanol, which was used as a case study for the volatile compound, in the biological samples. The limit of detection (LoD) was found to be 1.00 μL/mL, while at the same time evaporation of ethanol was prevented.

## 1. Introduction

It is commonly accepted that consumption of alcoholic beverages and drinks (wine, beer, etc.) helps human interaction, socialization, and euphoria. However, excessive use may have undesired or even dangerous consequences. Alcohol’s effects on humans vary depending on the levels in the body. When the alcohol concentration in blood is up to 1.27 μL/mL, it simply helps people to relax, but, when it exceeds this level, symptoms like car driving impairment appear. Intoxication is observed at levels above 2.53 μL/mL, while alcohol greater than 5.07 μL/mL in blood may lead to death [1]. There are multiple studies for the identification of alcohol also in other body fluids, such as saliva and urine. However, the intoxication levels are different. The ratio of alcohol in urine to blood was found approximately equal to 1.3, but it may be also enhanced according to the time of alcohol consumption and the amount of urine in the bladder [2,3,4,5].

The identification of volatile compounds in human body fluids could be very challenging. Concerning alcohol, the employed method is important to be suitable for analyte detection at least at the highest acceptable concentration for driving (1.27 μL/mL in blood and 1.65 μL/mL in urine). Among the established methods for determining the concentration of ethanol in a body is the breath alcohol test, which measures the concentration of ethanol in one’s breath using the breathalyzers [6]. The accuracy and precision of the breathalyzers, however, is controversial, as there are studies indicating uncertainty in the estimation of breath alcohol concentrations due to lack of sensitivity and specificity of the method [7,8]. For higher accuracy and precision, ethanol is determined in human blood through various types of gas chromatography (GC), including direct injection GC and headspace gas chromatography (HS-GC) [9]. The GC methods, also, can achieve very low limits of detection (LoDs) of alcohol in blood reaching as low as 10 μg/mL or 12.5 nL/mL [10] while using rather low quantity (100–500 μL) of the biological sample [10,11,12]. Although the GC methods are the most reliable in measuring the alcohol levels in human body fluids, they often suffer from sample preparation issues [13]. Moreover, GC is a destructive technique for the sample, while experienced personnel are required for the preparation of the samples and their analysis.

In order to overcome these issues, vibrational spectroscopies could be employed for determining alcohol levels in individuals’ bodies. Raman spectroscopy and infrared (IR) spectroscopy offer a rapid, easy in use, environmentally friendly, non-destructive, and non-invasive solution in biochemical analysis. The basic principle of Raman spectroscopy is based on the inelastic scattering of radiation by the examined sample [14]. The sample is irradiated by a monochromatic light source. Most of the photons are scattered by the sample at the same wavelength as the incident photons, a process known as Rayleigh scattering. Nevertheless, there are a few photons which are scattered at a shifted wavelength from that of the incoming radiation, which is referred to as inelastic scattering [15]. Inelastic scattering or Raman scattering, which is named after C.V. Raman, who discovered this phenomenon in 1928 [16], can be further divided into Stokes and anti-Stokes scattering [15], depending on the initial and the final vibrational state of the molecules.

Recently, Raman spectroscopy and other vibrational spectroscopic techniques have been employed for the analysis of body fluids, such as blood, serum, plasma, urine, saliva, semen, or vaginal fluids [17]. A variety of applications of Raman spectroscopy are described in literature. Raman spectroscopy has gained ground in the characterization of bone malfunctions, different types of cancer and other diseases, counterfeit drugs in bottles, as well as in the quantification of pharmaceuticals and their polymorphs in solid [18,19] and liquid formulations [20]. Among the applications of Raman spectroscopy, the determination of volatile essential and vegetable oils is included [21], while the identification of ethanol and methanol in alcoholic beverages [22] is also noteworthy. Lately, Raman spectroscopy was used for the determination of alcohol levels in the blood of individuals who consumed high quantities of ethanolic drinks [23]. However, there are still obstacles in the qualitative and quantitative analysis of such volatile compounds in biological fluids through Raman spectroscopy that should be surmounted.

Although the required sample preparation for Raman spectroscopy is quick and easy and the technique is sensitive for many biochemical applications, Raman scattering can prove inefficient because of the used substrate or sample holder [24]. The application of the wrong substrate can result in lower intensities of the studied sample or in fluorescent effects. A variety of different substrates and holders, as well as different sample preparation methods, have been used for studying biological fluids and volatile compounds via Raman spectroscopy. The most popular approach in Raman analysis of body fluids is the placement of a drop on a glass microscope slide which has been covered with aluminum foil in order to reduce the fluorescence from glass [25,26,27,28,29,30,31,32]. Another substrate that has been commonly used is a glass circular slide, which is designed appropriately so that it could be used with the mapping stage at the same time [33,34]. CaF_2_ has been also employed as substrate for the Raman analysis of variable biological fluids, such as serum and blood [35,36], while the application of silicon wafer surfaces [31] and silicon nanocrystal substrates has also been tested [37,38]. For forensic purposes, Raman spectroscopy for the analysis of bloodstains or other biological fluids has been applied on more sophisticated substrates, such as fabric clothes, denim from blue jeans, un-dyed cotton swatches, glass, metallic surfaces, walls, and a pale-yellow bathroom tile [39,40]. In most of the cases, the body fluids’ drops had been allowed to dry before their Raman spectra were acquired [25,26,27,28,29,30,33,34,40]. Concerning the Raman spectroscopic analysis of liquids and other fluid compounds, glass or quartz cuvettes [36,41,42,43] and well-sealed glass vessels [44] have been mentioned as sample holders in a couple of studies. The usage of such substrates could be demonstrated to be beneficial for the effective analysis of volatile compounds, such as ethanol, methanol, acetone, or hydrogen peroxide [42]. In addition, Raman spectra of liquid drug samples have also been acquired directly from their vials, ampoules, or bottles using a fiber optic probe [43,45].

Despite the benefits of the numerous different substrates for Raman spectroscopy that have been tested for biological and chemical applications, they are characterized by some weaknesses when volatile compounds in biological fluids are analyzed. The risks of volatility of these compounds, as well as the background noise due to the substrate, should be taken into consideration when the most appropriate substrate or sample holder carrier for Raman analysis of volatile compounds is chosen. Moreover, the required volume of the sample should be considered in the selection of the sample preparation method so that it will be at least comparable to the volume required in GC methods. At the same time, the developed method should certify a LoD below the highest acceptable levels by national laws.

In the present study, different substrates and sample holders have been tested, and the respective sample preparation methods have been developed for the identification of volatile compounds in body fluids. More specifically, the effectiveness of a glass microscope slide coated with gold layer creating a highly reflective carrier, a commercially available quartz cell for Raman analysis with a mirror on the back side, and a home-designed glass microscope slide with a cavity coated with gold layer, in the determination of ethanol in human urine, have been examined. Furthermore, the advantages and disadvantages of each carrier and sample holder have been studied.

## 2. Results

### 2.1. Identification of the Characteristic Peaks of Ethanol and Urine

The Raman spectra of ethanol, urine, and a sample of 5 μL/mL ethanol in human urine were recorded. The spectra of the biological fluids were acquired from a droplet of each sample which was placed on a highly reflective slide (glass microscope slide coated with gold substrate). However, the Raman spectrum of the volatile alcohol could not be recorded from a droplet, as it evaporated before the spectrum acquisition was completed. For this reason, the spectrum of ethanol was obtained using a commercially available cuvette for Raman spectroscopy.

The most prominent peak of urine was observed at 1003 cm^−1^ (Figure 1a), which is attributed to the symmetric C–N stretching of urea [46]. Although for solid urea the symmetric C–N stretching vibration is detected at 1010 cm^−1^, in solution it is shifted to lower wavenumbers [47]. Concerning ethanol, the most intense peak at 883 cm^−1^ is due to the symmetric stretching of the C–C bond. The Raman spectrum of ethanol is also characterized by less intense peaks at 434 cm^−1^ (bending vibration of C–C–O), 1053 cm^−1^ (asymmetric stretching of the C–O bond), 1096 cm^−1^ (rock vibrations of CH_3_), 1277 cm^−1^ (torsion and rotational vibrations of CH_2_), 1455 cm^−1^ (bending vibrations of CH_3_ and CH_2_), and 1482 cm^−1^ (bending vibrations of CH_3_) (Figure 1a) [48].

In the Raman spectrum of the 5 μL/mL ethanolic sample in human urine, the peak of urea at 1003 cm^−1^ was clearly detected, as well as the most prominent peak of ethanol. However, the peak of ethanol, which is attributed to the symmetric stretching of the C–C bond, was shifted to 880 cm^−1^ from the initial 883 cm^−1^. Similarly, the peaks of ethanol at 1053 cm^−1^ and 1096 cm^−1^ were shifted to 1048 cm^−1^ and 1089 cm^−1^, respectively (Figure 1b). These slight shifts to lower wavenumbers can be explained by the intermolecular interactions of ethanol and water. The molecules of water form a three-dimensional grid of hydrogen bonds among the hydroxylic groups. However, the addition of ethanol in concentrations up to 15–20% *w*/*w* leads to a structural rearrangement of the hydrogen bond network, as one molecule of ethanol is inserted for every five molecules of water. In this case, the hydrogen bonds are stronger than in water, resulting in weakening of the neighboring C–C and C–H bonds [49]. This is due to charge transfer leading to significant changes in the polarizability of the bonds. This change in polarizability is depicted in the Raman spectrum as a red shift, i.e., the peaks corresponding to the vibrations of the specific bonds are shifted to lower wavenumbers [48,49].

### 2.2. Method of Droplet on a Gold-Coated Glass Microscope Slide

The aim of this study was to develop a simple method requiring a minimum amount of sample (a few μL) and minimum sample preparation and offering low LoD of the analyte in the biological samples, while evaporation of the volatile compounds would be avoided. For this purpose, the determination of 100 μL/mL ethanol in human urine was used. The first method considered was that of placing a sample droplet on a highly reflective carrier (droplet method). The volume required was approximately 15 μL, and a gold-coated glass microscope slide was selected as a highly reflective sample carrier.

#### 2.2.1. Focus Optimization

The Raman spectra acquired from the droplet showed a dependency on the spot of the droplet on which the Raman laser beam was focused. Thus, three different spots of the droplet were investigated; the first (A) was on the side; the second (B) was in the middle and the third one (C) on the top of the droplet (Figure 2a). The Raman spectra acquired from the three different positions exhibited a deviation of the relative intensities of the two analytes (Figure 2b). This can be attributed to the absence of uniformity concerning the concentrations of the analytes in the droplet.

The characteristic peak of urine at 1003 cm^−1^ was easily detectable in all three Raman spectra. However, significant variations of the intensity of the most prominent peak of ethanol at 880 cm^−1^ were observed depending on the spot of the droplet on which the Raman laser beam was focused. More specifically, the peak of ethanol was barely distinguished from noise level in the Raman spectrum of the droplet’s spot (A), while the intensity of the ethanol peak is the maximum possible in the Raman spectrum of the spot (C) (Figure 2b). Thus, the spot (C) of the droplet is the optimum position to focus when the Raman spectrum is acquired, as the ratio of the peak intensities (ethanol to urine) is maximum. This is the outcome of the non-uniformity of ethanol concentration inside the droplet and can be attributed to mass and thermal convection phenomena [50]. Higher evaporation flux near the contact line leads to minimal ethanol concentration at the edge of the droplet (spot A). On the contrary, many ethanol molecules are trapped in the center near the top of the droplet where concentration is maximized [50].

#### 2.2.2. LoD Determination

For the LoD determination of the volatile compound in the droplet method, samples of human urine spiked with 0.25 μL/mL, 0.50 μL/mL, 2.00 μL/mL, 3.50 μL/mL, and 5.00 μL/mL ethanol were prepared. The Raman spectra of these samples were recorded after placing a droplet of each sample on a highly reflective gold-coated glass slide. The LoD of ethanol in human urine was determined, subsequently, by visual evaluation and signal-to-noise ratio as well [51]. The ethanolic peak at 880 cm^−1^ could not be detected in the 0.25 μL/mL sample Raman spectrum, but it was barely detected in the spectrum of the 0.50 μL/mL sample and easily detected in the spectrum of the 2.00 μL/mL sample (Figure 3). The signal-to-noise ratio was indeed higher than 3:1 [51] in the 0.50 μL/mL urine sample spiked with ethanol, while its value was below the minimum acceptable value (3:1) for the determination of the LoD (Table 1). Therefore, the LoD of ethanol in human urine when using the droplet method was found to be approximately 0.50 μL/mL.

#### 2.2.3. Kinetic Study of Ethanol Evaporation

The kinetics of ethanol evaporation from the droplet was investigated in a sample of 5 μL/mL ethanol in human urine. The Raman laser was focused on the top of the droplet (spot (C)), and sequential Raman spectra of the droplet were recorded. The time of each measurement was 57 s. The intensity of the ethanol peak was reduced with time. In the last Raman spectrum recorded after 10 min and 49 s since the placement of the droplet on the carrier, the intensity of the ethanolic peak was half of the initial one (Appendix A). The reduction rate of the ethanolic peak at 880 cm^−1^ with respect to the peak of urine at 1003 cm^−1^ was calculated for each Raman spectrum (0 min and 0 s, 0 min and 57 s, 1 min and 50 s, and 10 min and 49 s) (Appendix A). These results suggest that the evaporation of ethanol in the method of the droplet cannot be ignored despite the reduction in the scan time for the acquisition of the Raman spectra. Therefore, the rapid evaporation of the volatile compound appears to be the major drawback of the method, which outweighs the advantage of the small sample volume required and will result in great quantitative errors.

### 2.3. Method of Cuvette

As ethanol volatility proved to be a major obstacle in analysis in the previous section, another method which would prevent evaporation was sought. For this purpose, a commercially available cuvette for Raman spectra acquisition was employed. This cuvette was quite similar to the UV-Vis cells. It was a 5 mm thick synthetic quartz cuvette with a mirror on its back side so that the scattered radiation would be reflected for signal enhancement. The cuvette was fully filled with the appropriate volume of the sample (1.75 mL), plugged with a lid, and shielded with Parafilm M^®^ so as to exclude the air above the sample and inhibit evaporation.

#### 2.3.1. Focus Optimization

The optimal focus of the Raman laser beam on the cuvette was investigated. Five different positions were tested: the front side of the cuvette (A), various positions in the interior of the cuvette towards the back side (B)–(D), and the back side of the cuvette on the mirror (E) (Figure 4a). Due to the fact that the cuvette was not easily cleaned of the biological fluid, Xerostom^®^ was used instead for the study. Thus, the Raman spectra of Xerostom^®^ from the five different spots of the cuvette as well as its Raman spectrum, using the method of droplet, were recorded (Figure 4b). In the Raman spectra acquired from the front (A) and back (E) side of the cuvette, the most prominent peaks of the contents of the sample were absent. Although the peaks of Xerostom^®^ were detected in the Raman spectrum from the spot (B), their intensities were rather low. The Raman spectra from the positions (C) and (D) resulted in higher intensities of the characteristic peaks of the sample due to the longer distance of the radiation in the sample. The intensity ratio of the peaks in the Raman spectra of spots (B), (C), and (D) are practically identical to each other (Figure 4). However, the spots with the highest peak intensities are preferred, as lower limits of detection (LoDs) are expected in these cases. Therefore, focusing on the interior of the commercially available cuvette near its back side (spot (D)) is suggested for the Raman spectra acquisition when the method of cuvette is employed.

#### 2.3.2. LoD Determination

The ethanolic spiked samples of human urine (0.25 μL/mL, 0.50 μL/mL, 2.00 μL/mL, 3.50 μL/mL, and 5.00 μL/mL) were also analyzed using the method of cuvette so that the LoD of this method would be determined. After visual evaluation [40] of the respective Raman spectra, the LoD of ethanol in urine was determined to be equal to 2.00 μL/mL, as ethanol could not be detected in lower concentrations (Figure 5). Except for the visual evaluation, the LoD was also determined based on the signal-to-noise ratio method [51]. This method resulted in the same LoD as the one determined by the visual evaluation (Table 2). Thus, the method of cuvette results in a rather high LoD of ethanol in the specific body fluid.

#### 2.3.3. Kinetic Study of Ethanol Evaporation

The rate of ethanol evaporation from the sample in the cuvette was studied. A sample of 5 μL/mL ethanol in human urine was used for this study as in the method of the droplet. The Raman laser was focused on the interior of the cuvette close to the mirror on the back side (spot (D)), and sequential Raman spectra of the sample were acquired from the moment the sample was inserted and shielded in the cuvette. Each scan’s duration was adjusted to 5 min and 17 s. Only a slight decrease in the intensity of the peak of ethanol with time was observed (Appendix A). The rate of change of the height of the peak of ethanol at 880 cm^−1^ with respect to the height of the peak of urine at 1003 cm^−1^ was determined for four different time intervals (0 min and 0 s, 5 min and 17 s, 15 min and 52 s, and 31 min and 42 s). After approximately half an hour, the reduction of ethanol’s concentration was slightly higher than 12% (Appendix A). On the contrary, when the method of the droplet was applied, the concentration of ethanol was reduced to half after 10 min (Appendix A). Consequently, for the method of cuvette, the evaporation of the volatile constituent is obstructed to a large extent, but the significant quantity of the sample needed (1.75 mL) accompanied by the rather high LoD and the difficult cleaning of the biological fluid remain the most serious handicaps.

### 2.4. Method of a Gold-Coated Glass Slide with Cavity

#### 2.4.1. Gold-Coated Glass Slide with Cavity

In order to overcome the issues of large amount of sample of the cuvette method and the volatility of ethanol in the droplet method, a novel carrier was designed and constructed. More specifically, a glass microscopy slide with a cavity in the center was adopted. The slide, as well as the cavity, was coated with a gold substrate (EMF Corporation, Ithaca, NY, USA) offering high reflectivity to the carrier (Figure 6a). The volatile sample was deposited in the cavity. A quantity of 150 μL was required for complete filling of the cavity. An appropriate cover was also necessary to prevent evaporation of the volatile constituent which would allow the laser radiation to pass through at the same time (Figure 6b). Various transparent media have been tested and quoted below.

#### 2.4.2. Cavity Covering: Microscope Cover Slip

The first transparent medium tested for covering the sample was the microscope cover slip. A sample of 5 μL/mL ethanol in human urine was deposited in the cavity of the slide and covered with the glass cover slip. The laser was focused on the cover slip, and a Raman spectrum was collected which appeared identical to the one acquired when a sample’s droplet was caged between the gold coated glass slide and the cover slip. The peak of urine at 1003 cm^−1^ was barely detected, whereas no characteristic peak of ethanol could be observed (Figure 7). When the laser was focused in the main volume of the sample (under the cover slip), the peak of urea at 1003 cm^−1^ was clearly detectable; however, the most prominent peak of alcohol at 880 cm^−1^ could not be observed (Figure 7). Last, the cover was removed, and the spectrum of the sample was recorded. Both peaks of the urine at 1003 cm^−1^ and ethanol at 880 cm^−1^ were clearly detected (Figure 7). It can be, thus, concluded that the glass microscope cover slip may protect the sample from evaporation, but no clear signal from the sample is collected. The minor constituent is not detected.

#### 2.4.3. Cavity Covering: Transparent Membrane

Since the glass of the microscope cover slip had a significant effect on the acquired Raman spectrum, a transparent membrane (food cling film) was tested as an alternative medium for covering the cavity.

The material of cling films is crucial. It varies among the different suppliers as well. Two different cling films were tested; the first one was a 0.006 mm thick poly(vinyl chloride-vinyl acetate-vinyl alcohol) transparent membrane (Sanitas^®^), while the second one was a 0.006 mm thick polyethylene low-density transparent cling film (Vileda Freshmate^®^). The Raman spectra of the two membranes were recorded and found to differ significantly. In the spectrum of the Sanitas^®^ transparent membrane only a few peaks were observed, none of them in the area between 880 cm^−1^ and 1003 cm^−1^. However, multiple humps were observed; a broad significant hump was indeed observed at 883 cm^−1^, which interferes with the peak of ethanol (Figure 8a). On the contrary, in the Vileda Freshmate^®^ film’s Raman spectrum a few intense peaks appeared in the spectral area 1060–1500 cm^−1^. None of them would interfere with peaks of ethanol and urine at 883 cm^−1^ and 1003 cm^−1^, while only a broad peak of very low intensity was observed at 883 cm^−1^ (Figure 8a).

The effect of Vileda Freshmate^®^ cling film’s small broad peak at 883 cm^−1^ on the detection of the ethanol peak at the same Raman shift was further investigated. A sample of 5 μL/mL ethanol in human urine filled the cavity, and a piece of the membrane covered it, carefully taking care to be completely in touch with the sample so that no air is trapped in between. Except for the cavity, the membrane covered a surface of the gold-coated glass slide around the cavity in order to prevent leaking of the liquid sample out of the well. The Raman spectrum of the sample was recorded before and after covering with the transparent membrane. Both peaks of urine at 1003 cm^−1^ and ethanol at 880 cm^−1^ were clearly detectable, while the extra peaks of the membrane did not affect the determination of either the alcohol or the biological fluid (Figure 8b). The peaks of urine and ethanol were as distinct as in the Raman spectrum of the sample recorded without the transparent membrane despite the presence of an additional small peak at 1063 cm^−1^ corresponding to the membrane (Figure 8b). No interference from the small and broad peak of the transparent membrane at 883 cm^−1^ was detected in the Raman spectrum of the sample, which was covered with the membrane (Figure 8b). Therefore, the Vileda Freshmate^®^ cling film was selected as the most appropriate transparent coating of the cavity.

#### 2.4.4. Focus Optimization 

Different laser focusing spots were investigated. The first was adjusted on the surface of the transparent cling film (A), the second in the sample just under the membrane (B), and the third one targeted the main volume of the liquid sample (C) (Figure 9a). The Raman spectrum recorded from the spot (A) revealed the significant effect of the membrane material on the spectrum. However, the peaks of ethanol at 880 cm^−1^ and urine at 1003 cm^−1^ were also detected (Figure 9b). Focusing under the membrane (B) resulted in a Raman spectrum in which the effect of the membrane on the spectrum was reduced significantly, while the signal from the peaks of both analytes was increased to a great extent (Figure 9b). When the main volume of the sample was focused (C), the peaks of the membrane were barely detectable; only the peak at 1063 cm^−1^ could be distinguished from noise level. On the contrary, the intensities of the peaks of ethanol at 880 cm^−1^ and urine at 1003 cm^−1^ reached maximum values compared to (A) and (B) (Figure 9b). Thus, the peaks of the Raman spectrum of Vileda Freshmate^®^ cling film did not overlap the peaks of ethanol and urine and did not obstruct the detection of both constituents.

#### 2.4.5. LoD Determination

The LoD of ethanol in urine was also determined for the method of the gold-coated glass slide with cavity. For this purpose, human urine sample was spiked with 0.30 μL/mL, 0.50 μL/mL, 1.00 μL/mL, 2.00 μL/mL, 3.50 μL/mL, and 5.00 μL/mL ethanol. The Raman spectra of these samples were recorded after placing them in the cavity of the gold substrate carrier and after covering the fluids with the Vileda Freshmate^®^ transparent membrane. The LoD of ethanol in human urine was found 1.00 μL/mL by visual evaluation (Figure 10), as this was the minimum concentration at which the analyte could be distinguished from noise level [40]. Moreover, the LoD of this method was found equal to 1.00 μL/mL, even when the method of signal-to-noise ratio was used, as this was the minimum concentration at which the signal-to-noise ratio was higher than 3:1 (Table 3) [51]. Thus, the determined LoD for the gold-coated glass slide with cavity method was found between the LoDs estimated for the droplet method and the cuvette method.

#### 2.4.6. Kinetic Study of Ethanol Evaporation

In order to study the kinetics of ethanol evaporation from the sample, the Raman laser was focused on spot (C), i.e., under the membrane and in the main volume of the sample in the cavity. Sequential Raman spectra were acquired. The duration of each scan was set at 5 min and 16 s. The evaporation rate of the alcohol from the 5 μL/mL ethanolic sample in human urine was found to be prevented significantly when the sample, placed in the cavity of the novel highly reflective carrier, was covered with the transparent cling film (Appendix A) in comparison with the evaporation rate of the ethanol of an identical sample placed as a droplet on a gold substrate flat carrier. More specifically, the rate of change of the height of the ethanol peak at 880 cm^−1^, with respect to the height of the urine peak at 1003 cm^−1^, was calculated for four different time intervals (0 min and 0 s, 5 min and 16 s, 15 min and 48 s, and 31 min and 43 s) (Appendix A). These rates of ethanol evaporation in the specific methodology are slightly elevated compared to the reduction rates of ethanol in the method of the cuvette (Appendix A); however, they are dramatically lower than the respective rates in the method of the droplet (Appendix A).

It is, thus, proved that the proposed carrier, consisting of a gold-coated glass microscope slide with a cavity of 150 μL volume covered with a cling film of appropriate material, meets all the required conditions set. The sample volume necessary is minimized; evaporation of volatile constituents is well prevented; it is simple, not expensive, and easily constructed while offering an acceptable LoD of the analyte.

## 3. Discussion

Raman spectroscopy offers a novel perspective in the characterization of biological fluids. Multiple applications of Raman spectroscopy for the biochemical analysis and identification of compounds and substances in body fluids could be enumerated substituting the officially established methods [15,17,18,23,24,25,26,27,28,29,30,31,32,33,34,35,36,39,40,46]. However, most of the methods described in these studies involve the drying of the sample of biological fluid prior the acquisition of the Raman spectra, and, consequently, much time is consumed for the sample preparation. The dried sample also suffers from a lack of homogeneity, because the hydrophilic substances are separated from the hydrophobic constituents during the drying process. [25,26,27,28,29,30,33,34,40]. On the contrary, in our study all three proposed methodologies make use of the biological samples without any further preparation, reducing the required time of analysis and eliminating issues of sample homogeneity. Moreover, the proposed methods should offer simple, quick, less expensive solutions with LoDs lower than the maximum acceptable alcohol levels in drivers [1], requiring smaller amounts of biological sample for the detection of volatile compounds in body fluids than the official GC methods [10,11,12] while preventing the evaporation of the volatile constituents at the same time.

Each one of the three methods described in this study, the method of droplet, the method of cuvette, and the method of gold-coated glass slide with cavity, has its own advantages and drawbacks (Table 4). The method of droplet on a microscope slide coated with gold is the simplest method among those three, requiring only a few seconds for preparation of the sample. Then, the Raman spectrum of the sample is acquired from the droplet, which is placed on the highly reflective carrier, completing the analysis in a few minutes. In addition, no special equipment is required, and the production of gold-coated glass slides is quite economical, while only a few μL (<15 μL) of sample are required for the analysis. The LoD of the volatile alcohol was rather low as well. This method would be ideal for non-volatile samples of small available volume. Nevertheless, when the aim of analysis is the identification of volatile compounds, such as ethanol, in body fluids, the method of droplet is not suitable, because it suffers from significant errors and repeatability issues. These problems are due to the rapid evaporation of great amounts of the volatile compound from the droplet before the acquisition of the Raman spectrum is completed. When the method of droplet was applied on a 5 μL/mL ethanolic sample in urine, more than 15% and more than half of the ethanol was evaporated in approximately 2 and 10 min, respectively (Figure 11).

The method of cuvette offered a solution in reducing the evaporation of the alcohol from the sample significantly. More specifically, only 5.42% of ethanol was evaporated after 15 min and 52 s and 12.28% after 31 min and 42 s from the moment the 5 μL/mL ethanolic sample in urine was added in the commercially available Raman cuvette with a mirror on the back side (Figure 11). This very slow rate of evaporation of the volatile compounds suggests a useful method for the characterization of such volatile substances in biological fluids. However, the glass of the cuvette had some effect on the signal of the acquired Raman spectra, increasing significantly the LoD of the volatile alcohol in human urine from 0.50 μL/mL (method of droplet) to 2.00 μL/mL (method of cuvette). Moreover, the volume of the required sample to fill the cuvette (1.75 mL) is significantly enhanced compared to a few μL required in the method of droplet. Even though a smaller cuvette, which is commercially available, is selected, the required volume of the sample is no less than 0.35 mL. Such great amounts of sample are not usually available, as far as the body fluids are concerned. Partially filled cuvettes did not result in prevention of the evaporation of the volatile compound. Another important drawback is the cost of the cuvettes for Raman spectroscopy, as they are far more expensive than the microscope slides coated with the gold substrate. The cleaning of the cuvettes also may be challenging and time-consuming, especially when biological fluids are used, because they are usually viscous and stick on the cuvette’s walls. Insufficient cleaning of the cuvette leads to impurities in the following samples, a decrease in the reflectiveness of the mirror adjusted to the back side of the cuvette, and, consequently, poor quality of the Raman spectrum.

The third method included the addition of the sample of biological fluid with the volatile compound in the cavity of a gold substrate carrier. The sample in the cavity was covered with a transparent cling film, which prevented the evaporation of the alcohol from a 5 μL/mL ethanolic sample in urine to a great extent. After 15 min and 48 s, only 15.44% of ethanol was evaporated, while after 31 min and 43 s approximately 21% of the total ethanol was evaporated (Figure 11). Even though the reduction rate of the volatile alcohol from the sample was higher than in the method of the cuvette sealed with the lid, it was still significantly lower than in the method of the uncovered droplet on the highly reflective carrier. In addition, the specific method was suitable for detecting the analytes below the transparent membrane, as its effect on the Raman spectrum was not as great as the effect of the glass in the case of the cuvette. However, in order to avoid the interference of the membrane in the Raman spectrum, a suitable membrane, whose peaks will not obstruct the detection of the analytes, should be selected, while at the same time the Raman laser should be focused on the main volume of the sample in the cavity and not just below the membrane. The highly reflective gold substrate of the slide assisted in the acquisition of high-quality Raman spectra. Furthermore, the LoD of ethanol in human urine when using the gold-coated glass slide with cavity method was estimated at 1.00 μL/mL. Although the LoD is higher than the one found in the method of droplet (0.50 μL/mL), it is half of the respective LoD determined for the method of cuvette (2.00 μL/mL). The estimated LoD of 1.00 μL/mL, however, is acceptable for identifying individuals intoxicated by ethanol, as light intoxication is observed for blood ethanol concentrations higher than 1.27 μL/mL, corresponding to a urine alcohol concentration of 1.65 μL/mL [52]. In addition, the determined LoD is lower than the highest acceptable limit of alcohol in urine when driving in most countries, which is 1.30 μL/mL [1]. Thus, it can be concluded that the proposed setting can be applied as a quick, easy, and non-destructive tool for on-site intoxication control before more quantitative official approaches are applied. Another advantage of this method was the relatively low volume of the required sample. Only 150 μL of the sample could completely fill the cavity. Although this volume was higher than the volume of the droplet, it was at least three-fold lower than the required volume in the method of the cuvette. In addition, compared to the GC methods that are officially employed for the determination of ethanol in biological samples, such as HS-GC, the volume of the sample required in the method of the gold-coated slide with cavity is equal to or less than the volume of biological sample required in HS-GC (100–500 μL) [10,11,12]. Moreover, the cleaning of the carrier was not as challenging as the cleaning of the cuvette. The sample was removed by washing the slide with ultra-pure water and ethanol after each usage. In order to dry the liquid drops after cleaning, warm air was preferred instead of soft tissue so that the gold substrate would not be destroyed. Finally, the cost of purchasing and coating with gold the microscope slide with the cavity was significantly lower than the cost of purchasing the commercially available cuvette for Raman spectroscopy.

At this point, it is important to mention that this is the first time in literature that such a carrier and sample preparation method for Raman spectroscopy was designed, proposed, and used for the identification of liquid samples, especially for volatile compounds in biological fluids. Furthermore, the coating of flat microscope slides with gold [26,53,54] and the usage of variable types of cuvettes for the Raman analysis of liquid samples [36,41,42,43,44] are referred to in previous studies; however, there is no reference in literature about using as substrate for Raman spectroscopy the cavity of a microscope slide which was coated with gold.

## 4. Materials and Methods

### 4.1. Samples

Absolute ethanol of analytical reagent grade (Fisher Scientific UK Ltd., Loughborough, UK) has been used for recording ethanol Raman spectrum and for preparing the mixtures with urine. Urine samples from three volunteers of the Laboratory of Instrumental Pharmaceutical Analysis, Department of Pharmacy, University of Patras, Greece were collected. Their age ranged from 20 to 55 years, and they were not suffering from any severe or infectious disease, such as cancer or AIDS. The urine samples were collected in a sterile urine collector. Permission for the implementation of this study was administered by the Bioethics Committee of University of Patras, Greece (Protocol Number: EB 45). No further processing of the urine samples was applied. All three volunteers have assured that they did not consume any alcoholic drinks for the previous 48 h prior the urine collection.

The 5 μL/mL ethanolic sample in human urine was prepared by adding the appropriate quantity of absolute ethanol in a volume of human urine. The final volume of each sample was 3 mL. For transferring the appropriate volume of absolute ethanol, a 10 μL automated pipette was used (Pipetman Neo^®^, Gilson Inc, Middleton, WI, USA), while a 1 mL or 5 mL automated pipette was employed for the human urine (BioPette^®^ Autoclavable Pipettes, Labnet International Inc, Edison, NJ, USA). The samples were homogenized by shaking in a vortex (MS2 Minishaker, IKA^®^-Werke GmbH & CO., KG, Staufen im Breisgau, Germany). The samples were analyzed immediately after their production, and, subsequently, they were stored at −20 °C until the next measurement.

Xerostom^®^ oral spray 15 mL (Pharmaserve-Lilly, Kifisia, Athens, Greece), which is used for the relief of dry mouth, was purchased from a local drugstore.

### 4.2. Raman Spectroscopy

The Raman spectra were acquired by using an InVia Raman spectrometer (Renishaw, Wotton-under-Edge, UK) coupled with optical microscope (DM Leica, Leica Microsystems, Wetzlar, Germany). The samples were excited with a 785 nm wavelength laser diode. The spectral resolution was 2 cm^−1^, and a charge-coupled device (CCD) detector was used. The nominal value of laser power was 250 mW, while the laser power on the edge of the microscope lens was measured at 32 mW. A 20x, 0.4 NA objective lens (model 566026, Leica Microsystems, Wetzlar, Germany) was employed for the measurements. The Raman spectra of ethanol, human urine, and the materials used as substrates were recorded in the spectral region of 100–2000 cm^−1^, while the Raman spectra of the ethanolic samples in human urine were acquired in the spectral region of 700–1200 cm^−1^. The spectra were acquired through the Windows-based software WiRE^©^ 2.0.

For the daily calibration of the spectrometer, the Raman spectrum of a silicon reference standard was obtained before the acquisition of the Raman spectra of the samples. When recording the silicon reference standard spectrum, the laser power was set at 5 × 10^−8^% of the maximum power, and the time of scan was adjusted to 1 s. Each spectrum of the reference standard was a result of two accumulated scans. The Raman shift of silicon peak at 520 cm^−1^ and its intensity were used to validate the calibration of the spectrometer.

#### 4.2.1. Raman Spectra Acquisition Using the Method of Droplet

The first method for the acquisition of Raman spectra involved the usage of a special highly reflective slide designed by coating a glass microscope slide with gold substrate (EMF Corporation, Ithaca, NY, USA). The coating of the slides was constituted by two layers; the first one was a layer of titanium (50 Å) binding the glass of the slide with the gold coating, while the external one was a bare gold layer (1000 Å). This gold layer offered a high reflection on the carrier. The dimensions of the slides were 2.6 × 7.6 cm, and their thickness was 1.0 mm. A droplet of each sample (approximately 5–10 μL) was placed on the carrier with the gold substrate using a 1 mL syringe. Movement of the point of focus to predetermined distances was achieved using the high precision levers of the microscope stage following focusing on the top of the droplet.

#### 4.2.2. Raman Spectra Acquisition Using the Method of Cuvette

A commercially available macro-cuvette 100-QX with a polytetrafluoroethylene (PTFE) lid (Hellma, Müllheim, Germany) with a mirror on the back side which contributed to the reflection of the scattered radiation was the second carrier used for the acquisition of the Raman spectra. A volume of approximately 1750 μL was necessary for filling completely the cuvette. The cuvette was created from synthetic quartz glass (Suprasil^®^ 300, Heraeus Quarzglas GmbH & Co., Hanau, Germany), and its path length was 5 mm. The height of the cuvette was 45 mm, the width 12.5 mm, and the depth 7.5 mm; its inside width was 9.5 mm, and its base thickness was 1.5 mm. The cuvette was filled with the sample completely, and the cap was placed carefully on top and covered with Parafilm M^®^ (Bemis Company, Inc., Neenah, WI, USA) in order to seal it tightly so that no air bubbles would be trapped in the sample. The Raman laser was focused on the sample in the cuvette at an angle of 90° by using an angle mirror lens. Movement of the point of focus to predetermined distances was achieved using the high precision levers of the microscope stage following focusing on the outer wall of the cuvette.

#### 4.2.3. Raman Spectra Acquisition Using the Method of Gold-Coated Glass Slide with Cavity

For the third proposed methodology, a microscope slide with a 1-well cavity was coated with gold (EMF Corporation, Ithaca, NY, USA) as the simple microscope slide in order to obtain a highly reflective carrier. Thus, a 50 Å titanium layer was deposited on the glass of the slide, and a 1000 Å bare gold layer was added as the external layer. The slides’ dimensions were 2.6 × 7.6 cm with 1.25 mm thickness. The diameter of the cavity was 1.5 cm, while the well’s depth was 0.6 mm. The sample was placed in the well, which required 150 μL to be filled. The cavity with the sample was covered using a piece of a 0.006 mm thick polyethylene low-density transparent cling film (Vileda Freshmate^®^, FHP Hellas, Kifisia, Athens, Greece) in order to prevent ethanol evaporation. A second 0.006 mm thick poly(vinyl chloride-vinyl acetate-vinyl alcohol) transparent membrane (Sanitas^®^, Sarantis S.A., Marousi, Greece) was also tested. Movement of the point of focus to predetermined distances was achieved using the high precision levers of the microscope stage following focusing on the cling film.

#### 4.2.4. Repeatability of the Method of Gold-Coated Glass Slide with Cavity

The repeatability of the method was tested using a human urine sample spiked with 5.00 μL/mL ethanol. Three consecutive measurements were performed. A fresh quantity of sample was used each time and placed in the cavity of the gold-coated glass slide and covered with cling film. The laser was focused on spot C, i.e., in the main volume of the sample. The cavity slide was cleaned and dried between measurements. The ratio of the intensity of the ethanol peak at 880 cm^−1^ to the intensity of the urine peak at 1003 cm^−1^ was calculated for each measurement. For the evaluation of the repeatability of the method, the relative standard deviation was determined (Table 5).

#### 4.2.5. Determination of the LoD

The LoD of ethanol in human urine was estimated by visual evaluation and signal-to-noise ratio for each of the three proposed methods [51]. Regarding, the method of droplet and the method of cuvette, five samples of human urine spiked with ethanol (0.25 μL/mL, 0.50 μL/mL, 2.00 μL/mL, 3.50 μL/mL, and 5.00 μL/mL) were prepared. The samples were homogenized by shaking in a vortex (MS2 Minishaker, IKA^®^-Werke GmbH & CO. KG, Staufen im Breisgau, Germany), and, subsequently, their Raman spectra were acquired. The same procedure was also followed for the determination of the LoD in the method of gold-coated glass slide with cavity. In this case, six mixtures of ethanol with human urine (0.30 μL/mL, 0.50 μL/mL, 1.00 μL/mL, 2.00 μL/mL, 3.50 μL/mL, and 5.00 μL/mL) were prepared and homogenized, and their Raman spectra were recorded.

#### 4.2.6. Study of the Kinetics of Ethanol Evaporation

For each of the three proposed methodologies, the rate of ethanol evaporation from a 5 μL/mL ethanolic sample in human urine was determined. For this purpose, the ratio of the intensity height of ethanol’s peak at 880 cm^−1^ to the intensity height of the urine’s peak at 1003 cm^−1^ was calculated for the initial measurement and for each time interval. For the determination of the evaporation rate of ethanol, the rate of change of the ratio at each time interval with respect to the initial ratio was determined according to the following equation:(1)$$\mathrm{Rate}\;\mathrm{of}\;\mathrm{Change}=\frac{\frac{I_{\mathrm{EtOH},t}}{I_{\mathrm{Urine},t}}-\frac{I_{\mathrm{EtOH},t0}}{I_{\mathrm{Urine},t0}}}{\frac{I_{\mathrm{EtOH},t0}}{I_{\mathrm{Urine},t0}}}\;\times \;100\%,$$
where I_EtOH,t_ and I_Urine,t_ are the intensities of ethanol’s peak at 880 cm^−1^ and urine’s peak at 1003 cm^−1^, respectively, at a specific time interval, while I_EtOH,t0_ and I_Urine,t0_ are the intensities of ethanol’s peak at 880 cm^−1^ and urine’s peak at 1003 cm^−1^, respectively, at the initial time point (t = 0 min and 0 s).

## 5. Conclusions

All three proposed methods in this study could be used for the acquisition of the Raman spectra of liquid samples. However, each one is limited by certain drawbacks and is suitable for certain types of sample. The usage of the gold-coated slide with the cavity, which is covered with a transparent cling film, demonstrated certain advantages over the other two methods for the identification of volatile liquid samples. Particularly, the use of the proposed methodology for the determination of other volatile compounds (ethanol, methanol, acetone, isopropyl alcohol, etc.) in variable body fluids (blood, serum, urine, saliva, semen, vaginal fluids, etc.) could be proved beneficial, as it is a simple, fast, inexpensive, user-friendly, and environmentally friendly method requiring only a few μL of sample, offering low LoDs and high protection from evaporation of the volatile constituents. Moreover, the proposed method could be used as a quick approach for detecting the alcohol levels in drivers who have consumed alcoholic beverages above the maximum acceptable limit. However, for the detection of small amounts of alcohol in human bodies, the conventional methods cannot be replaced. 

## Figures and Tables

**Figure 1 molecules-27-03279-f001:**
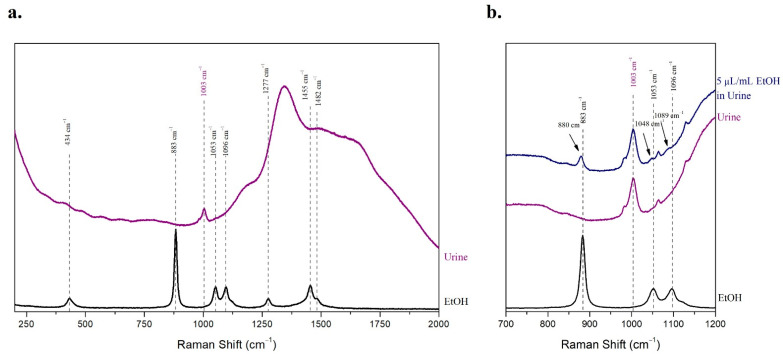
The Raman spectra of ethanol, human urine, and the 5 μL/mL ethanolic sample in human urine were acquired. (**a**) The characteristic peaks of ethanol and human urine are noted with dashed lines; (**b**) the characteristic peaks of ethanol were shifted in the sample of 5 μL/mL ethanol in human urine, and the shifted peaks are noted with arrows.

**Figure 2 molecules-27-03279-f002:**
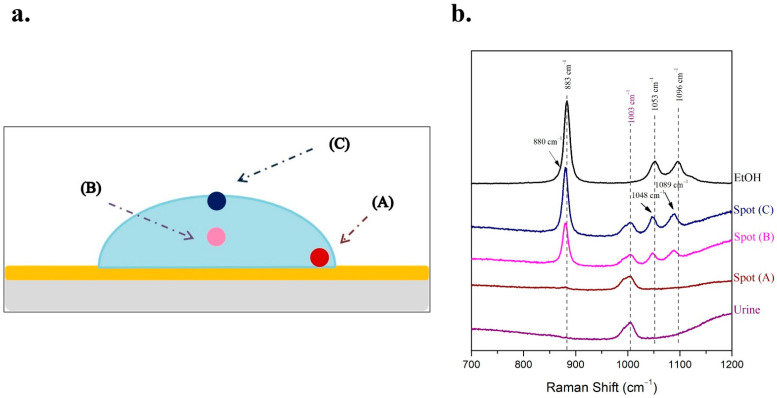
The method of sample droplet on a glass microscope slide coated with a gold highly reflective substrate was applied for the determination of 100 μL/mL ethanol in human urine. (**a**) Schematic illustration of the focus on three different spots of the sample droplet; (**b**) the respective Raman spectra from the three different spots of the droplet were recorded.

**Figure 3 molecules-27-03279-f003:**
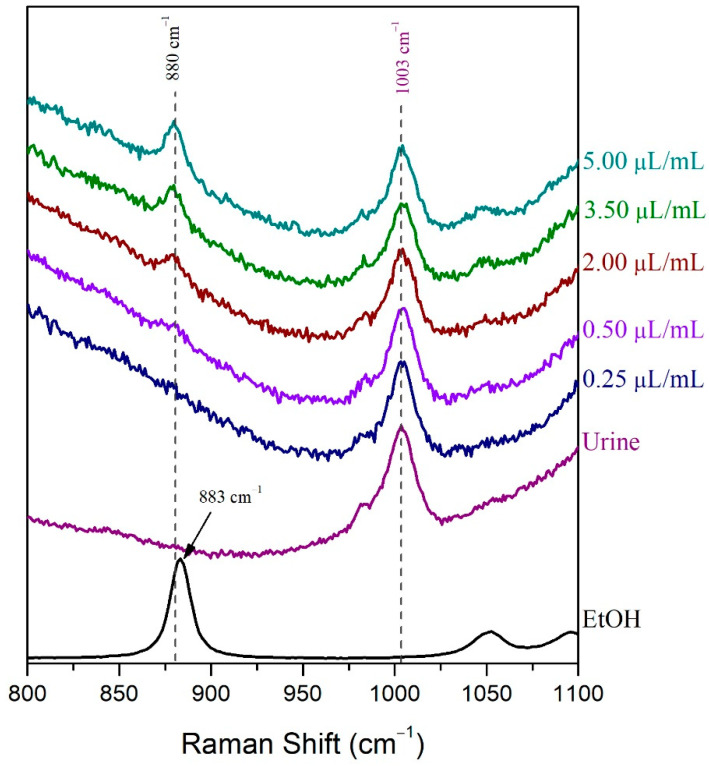
The LoD of ethanol in human urine using the droplet method was determined by visual evaluation.

**Figure 4 molecules-27-03279-f004:**
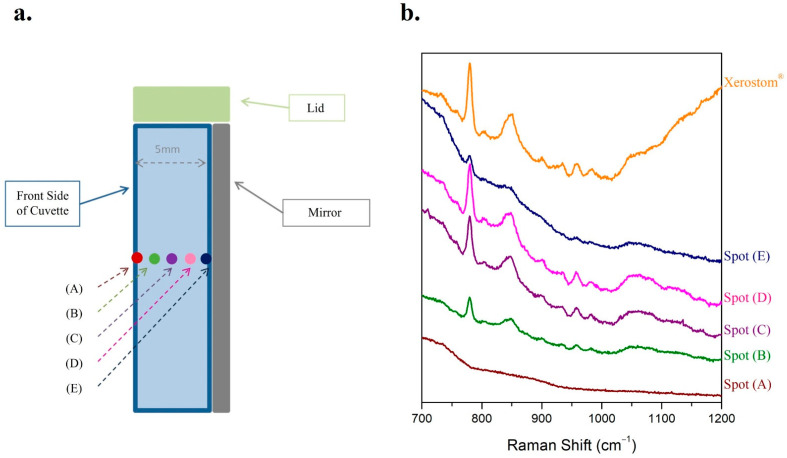
The method of the commercially available cuvette was applied on a liquid sample of Xerostom^®^. (**a**) Schematic illustration of the focus on five different spots of the cuvette; (**b**) the respective Raman spectra of Xerostom^®^ from the five different focuses on the cuvette were recorded.

**Figure 5 molecules-27-03279-f005:**
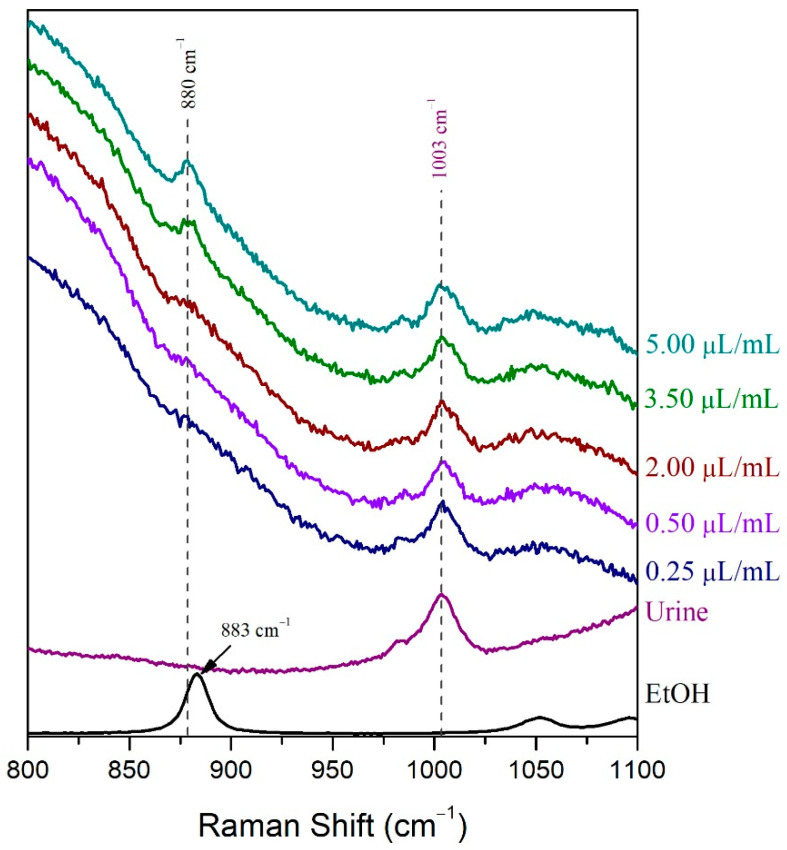
The LoD of ethanol in human urine using the method of cuvette was determined by visual evaluation.

**Figure 6 molecules-27-03279-f006:**
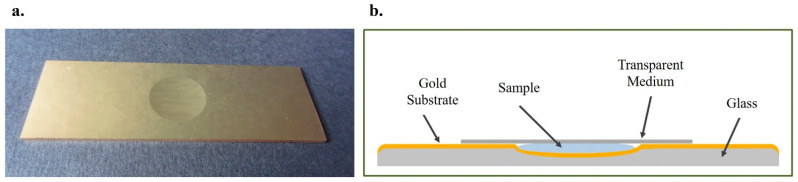
(**a**) The microscope slide with the cavity in the center of the slide coated with gold was used for recording the Raman spectra of volatile compounds in biological fluids; (**b**) schematic illustration of the glass microscope slide with the cavity coated with gold and covering the sample in the cavity with a transparent medium.

**Figure 7 molecules-27-03279-f007:**
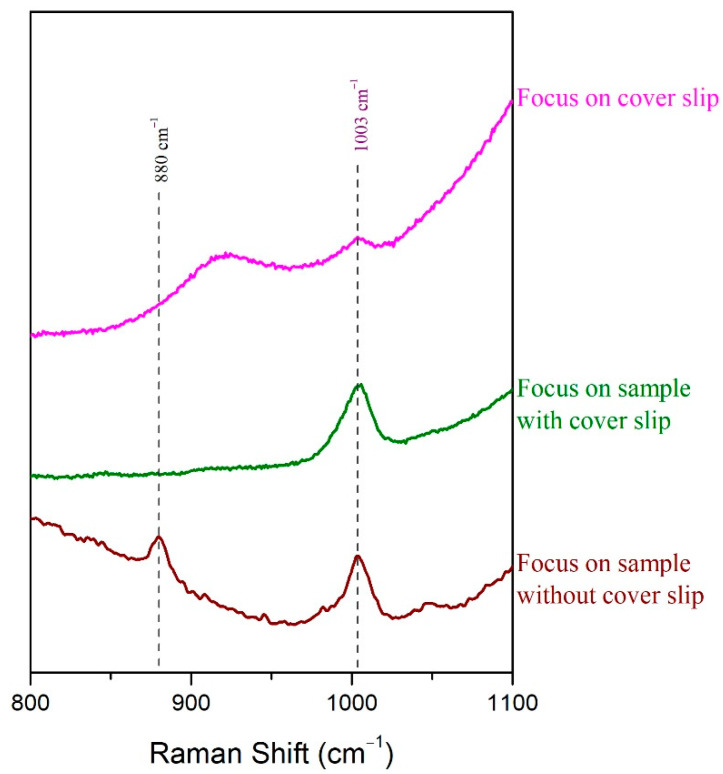
The Raman spectrum of 5 μL/mL ethanol in human urine after placing the sample in the cavity of the gold-coated glass slide was recorded without and with being covered with a microscope cover slip. Two different focus levels were tested when the cover slip was applied.

**Figure 8 molecules-27-03279-f008:**
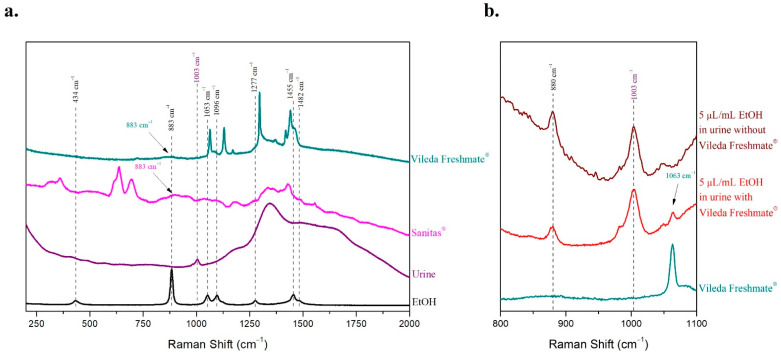
(**a**) The Raman spectra of ethanol, human urine, and the two transparent cling films (Sanitas^®^ and Vileda Freshmate^®^) were recorded, and the characteristic peaks of ethanol and urine are noted with dashed lines; (**b**) the Raman spectrum of the 5 μL/mL ethanol in human urine sample was recorded without and with being covered with the Vileda Freshmate^®^ membrane.

**Figure 9 molecules-27-03279-f009:**
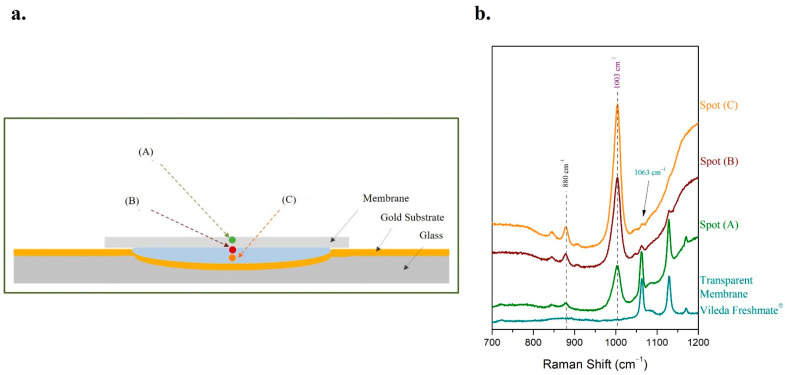
The method of placing the sample droplet in a cavity of a home-designed glass microscope slide coated with a gold highly reflective substrate and covering the sample with a piece of transparent cling film (Vileda Freshmate^®^) was applied for the determination of 5 μL/mL ethanol in human urine. (**a**) Schematic illustration of the focus on three different spots of the sample in the cavity; (**b**) the respective Raman spectra from the three different spots of the sample in the cavity were recorded.

**Figure 10 molecules-27-03279-f010:**
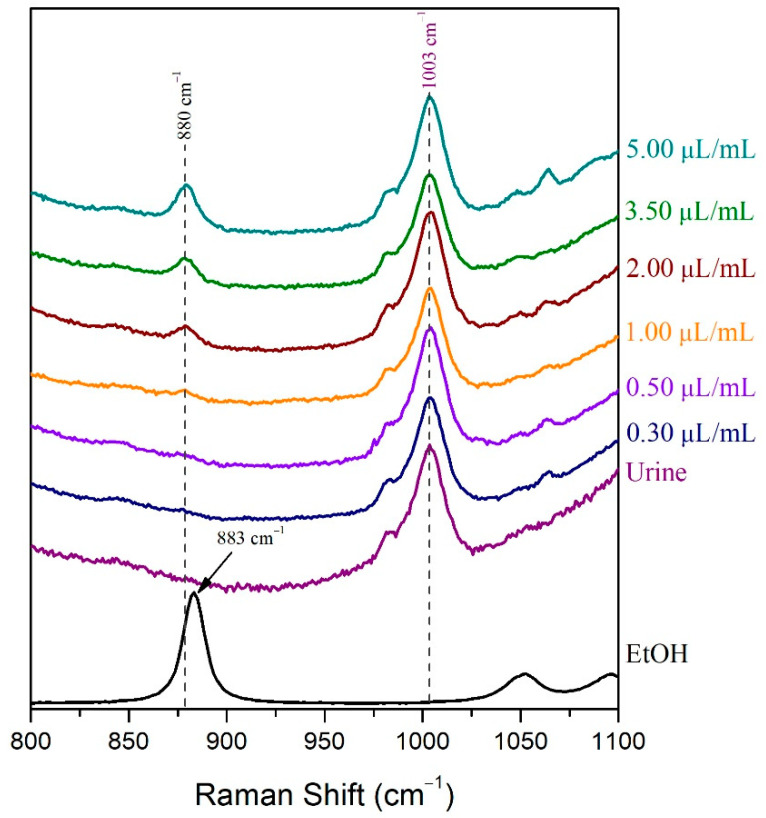
The LoD of ethanol in human urine using the method of gold-coated glass slide with cavity was determined by visual evaluation.

**Figure 11 molecules-27-03279-f011:**
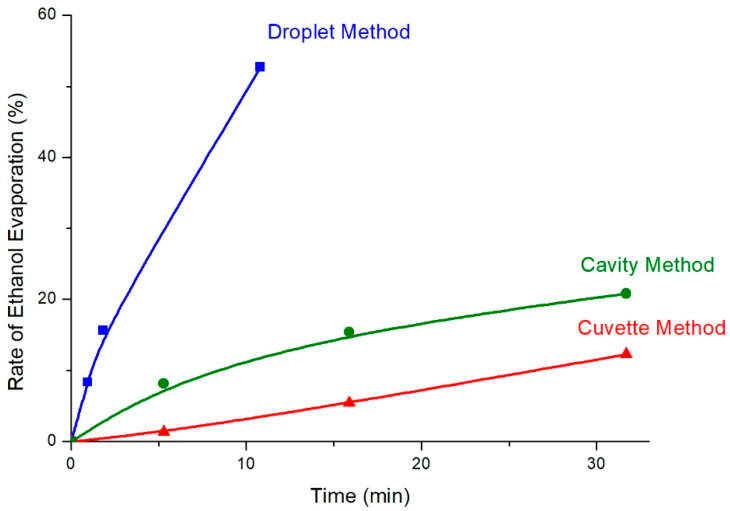
Rate of ethanol evaporation as a function of time for the method of droplet, the method of cuvette, and the method of the gold-coated glass slide with cavity.

**Table 1 molecules-27-03279-t001:** The LoD of ethanol in human urine for the droplet method was determined based on the signal-to-noise ratio.

Sample	Signal-to-Noise Ratio
Urine	1.00
0.25 μL/mL	2.40
0.50 μL/mL	3.62
2.00 μL/mL	4.99
3.50 μL/mL	8.49
5.00 μL/mL	11.52

**Table 2 molecules-27-03279-t002:** The LoD of ethanol in human urine for the method of cuvette was determined based on the signal-to-noise ratio.

Sample	Signal-to-Noise Ratio
Urine	1.00
0.25 μL/mL	2.70
0.50 μL/mL	2.90
2.00 μL/mL	6.72
3.50 μL/mL	10.95
5.00 μL/mL	14.20

**Table 3 molecules-27-03279-t003:** The LoD of ethanol in human urine for the method of gold-coated glass slide with cavity was determined based on the signal-to-noise ratio.

Sample	Signal-to-Noise Ratio
Urine	1.00
0.30 μL/mL	1.96
0.50 μL/mL	1.96
1.00 μL/mL	3.17
2.00 μL/mL	6.86
3.50 μL/mL	9.44
5.00 μL/mL	16.79

**Table 4 molecules-27-03279-t004:** The advantages and disadvantages of each method.

Factor	Method of Droplet	Method of Cuvette	Method of Cavity
Simplicity	The simplest	Very simple	Very simple
Required Time of Sample Preparation	A few seconds	Some seconds	Some seconds
Required Time of Analysis	A few minutes	Some minutes	Some minutes
Required Volume	<15 μL	350–1750 μL	150 μL
Cost	Low	High	Low
Cleaning	Easy	Difficult	Easy
Interference of Carrier’s Material on the Raman Spectrum	No effect	High interference of the glass	Very low interference of the transparent membrane
LoD	Very Low (0.50 μL/mL ethanol in urine)	High (2.00 μL/mL ethanol in urine)	Low (1.00 μL/mL ethanol in urine)
Evaporation Rate of Volatile Samples	Very fast	Very slow	Slow

**Table 5 molecules-27-03279-t005:** Repeatability of the method of gold-coated slide with cavity for a urine sample spiked with 5 μL/mL ethanol.

Measurement	I880/I1003
1st Measurement	0.317
2nd Measurement	0.313
3rd Measurement	0.303
Average	0.311
Standard Deviation	0.007
Relative Standard Deviation (%)	2.25

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
