# Peer review of "Comparative Study of Sample Carriers for the Identification of Volatile Compounds in Biological Fluids Using Raman Spectroscopy"

_molecules, 2022, doi:10.3390/molecules27103279_

Round 1

Reviewer 1 Report

In this study, three carriers are proposed and the respective sample preparation methods are described for the determination of ethanol in human urine samples. The advantages and disadvantages of each method were comprehensively compared, which would provide valuable information for the researcher working on biological detection using Raman spectroscopy. In general, this manuscript was well organized and presented. Some issues should be revised or clarified before final recommendation.

  1. In Figure 2, the signals from spot A was weaker than those from spots B and C. The substances in spot A were closer to the gold layer, which was supposed to have more and higher Raman signals due to the surface enhanced Raman scattering effect. Please explain it.
  2. The results showed that the Raman signals from different spots varied greatly. In real practice, how can operator obtain the repeatable signals from samples?
  3. The storage time of these carriers coated with gold layer should be evaluated.

Author Response

Reviewer 1

  1. In Figure 2, the signals from spot A was weaker than those from spots B and C. The substances in spot A were closer to the gold layer, which was supposed to have more and higher Raman signals due to the surface enhanced Raman scattering effect. Please explain it.

In Figure 2, the signal of urine (1003 cm-1) at spot A is almost the same as at spots B and C. Surface enhanced Raman scattering effect does not take place. The gold layer of the carrier acts only as a mirror i.e. has not been modified to act as SERS substrate.

The ethanol signal (880 cm-1) is maximum at spot C (top of the droplet) and is barely detected at spot A. This is described and commented in section 2.2.1 of the manuscript, “…This implies that ethanol is not equally distributed in the droplet and concentrates on the top before evaporation takes place…”. More explanation on this has also been added in the revised manuscript.

  1. The results showed that the Raman signals from different spots varied greatly. In real practice, how can operator obtain the repeatable signals from samples?

Laser focus is necessary before Raman spectrum recording, regardless of the type of the sample (solid or liquid). This is accomplished using the stage of the microscope on which the sample is positioned.

In this work and for the liquid sample carriers studied, movement of point of focus can be done using the high precision levers of the microscope stage after focusing on a defined surface, the front wall for the cuvette, the cling film for the cavity slide, the air-liquid interface for the droplet. The distances of focus point movement can be predetermined, repeatable and accurate.  

  1. The storage time of these carriers coated with gold layer should be evaluated.

The storage time of the gold coated glass slides is unlimited. They are not modified SERS substrates.

Reviewer 2 Report

The introduction and results sections do not provide sufficient quantitative measures or goals to evaluate the quality or utility of the results and proposed methods.  The goals of the project are described as obtaining “good” detection limits with a minimal quantity of sample.  It would be better to indicate quantitatively early in the manuscript what the required figures of merit are.  Descriptions of detection limits as “low” or “rather low” is not informative.  The relevant detection limit for ethanol in urine is not presented until the conclusions section of the manuscript.

The authors have overstated their case with respect to the limitations of the competitive headspace GC methods, describing them as “suffering from sample preparation issues” and being a “catastrophic technique” (it is assumed that they meant “destructive”), but these are not necessarily true limitations of GC or of the GC methods for the proposed analyses.

On page 2 the authors refer to the “initial and final vibrational state of the photons.”  The photons do not have vibrational states.

The authors describe quite different signal intensities as a function of the point of focus.  This generates concerns regarding reproducibility of the methods and approaches.  It is not clear that the authors have made measurements on multiple samples to determine the reproducibility or the effect of focus point on reproducibility. 

On page 5 the authors hypothesize that the difference in signal is caused by ethanol not being equally distributed in the drop and being concentrated at the surface before evaporation.  This seems unlikely, as evaporation is likely to occur from the surface leading, if anything, to a concentration gradient decreasing from the interior to the surface.  An alternative explanation could be surface activity of the ethanol, but that also seems unlikely.  Further, significant differences in signal intensity are observed as a function of focus point in the other sampling methods without reason to believe that there is a gradient in concentration.  Overall this is more likely to be an instrumental effect.

LODs are determined by “visual evaluation,” and a reference is provided to justify this approach.  The provided reference also recommends more quantitative statistical approaches to determining LOD.  The authors should consider and use a more quantitative approach to determining and reporting LODs.

The text describing of the rate of ethanol absorption for the various approaches is too lengthy and relatively confusing.  Figures 4, 7, and 14 that show signals at different time points are not particularly informative and could be eliminated.  Tables 1, 2 and 3 could also be eliminated and replaced with a single figure that shows signal as a function of time for the three different approaches, providing a visual image of the effect of evaporation.

The text regarding the results with a glass cover slip, including figures 8 and 10, is too lengthy and detailed.  It would suffice to say that the glass cover slip interfered with the detection of ethanol and could not be used.  Figures 8 and 10 could be included in supplemental information.

The conclusions section is repetitive with the results section and should be reduced in length by at least half.  The conclusions should summarize the results as in Table 4, but should not present all of the same results as already presented in the results section.

The discussion of LOD in the conclusions section leaves this reviewer concerned about the general utility and applicability of the methods.  It is true that a single measurement of LOD at 1.00 uL/mL is below the reported intoxication value of 1.27 uL/mL.  But application of this method in, for example, a forensic environment would require quantitation of the ethanol at concentrations at and below 1.27 uL/mL.  As 1.27 uL/mL is just above the LOD (which itself was only determined visually), it is not clear that any of these methods could provide the necessary level of uncertainty in quantitative analyses. 

The authors do not indicate if there is a per se limit for ethanol in urine when operating a motor vehicle, for example, or whether urine samples are commonly used in forensic applications.

The authors are ultimately pleased that their recommended system requires “only 150 uL” of sample.  But they do not compare this quantitatively to competitive methods.

Reviewer 3 Report

The article describes a comparative study of different sample preparation and presentation methods for the analysis of volatile compounds (ethanol) in biological fluids (human urine) using Raman Spectroscopy. While the study is sound, it is not very novel, and some aspects of the techniques are specific to the experimental conditiona used by the authors, and not very generalisable. Issues to be considered:

(i) An important consideration is, what range of concentrations and limit of detection is required/desired. Are there legislative guidelines? The LOD achieved (which should be quoted in the Abstract) should be presented in this context.

(ii) "The most popular approach in Raman analysis of body fluids is the placement of a drop on a glass microscope slide, which has been cov ered with aluminum foil in order to reduce the fluorescence from glass [17-22]."
At least 5, of not all 6 of these references come from the same group.  Reference 25 and 26, for example argue for measurement in a different way. Reference 26 does not use CaF2 substrates, however, as seems to be implied. Also, in 26, emphasis is not on "body fluids’ drops had been allowed to dry before their Raman spectra were acquired".

(iii) What is meant by "...as well as the low intensity due to the substrate". In general, the purpose/influence of a reflecting substrate on the intensity of the Raman signal collected, or its role in considering the optimisation process has not been discussed. The reflector can effectively double the signal copllected, by retroreflecting the forward scattered signal, but can also "fold" the laser focus, doubling the interaction with the sample, if the sample thickness is less than the focal depth of the microscope objective. This effect is therefore dependent on the magnification of the objective, which has not been varied.

(iv) as a continuation of the previous point, in the cuvette methid, the  interaction with the sample and therefore the intensity of the Raman collected depends on the focal depth, but there is also a trade off with the NA. In general, a x10 is optimum.

(v) "The most prominent peak of urine was observed at 1003 cm-1 (Figure 1a),..." the predominant constituent of urine is water, which has a features at 1640cm-1, and should dominate the spectrum. The spectrum of Figure 1(a) does not show this, and the authors need to check, for example what is the spectrum of the substrate on its own. Note that the spectrum of Figure 1(a) is very similar to that of glass in Figure 8, suggesting that the gold coating is not fully reflective.

(vi) In figure 2, the legends indicating which curve is which should be more prominant.

(vii) There are more appropriate (multivariate) methods for determination of LoD which should be used.  
Allegrini F, Olivieri AC (2014) IUPAC-consistent approach to the limit of detection in partial least-squares calibration. Anal Chem 86:7858–7866

(viii) The full spectrum should be shown for the cuvette method.

(ix) In general most of the full range spectra indicate the contribution of fluorescence of glass. this is an issure for the use of 785nm as lasr source, but not, for example for 532nm. this should be mentioned and the results duscussed in this context.

Reviewer 4 Report

This article “Comparative Study of Sample Preparation Methods and Carriers for the Identification of Volatile Compounds in Biological Fluids using Raman Spectroscopy” by Panagiota Papaspyridakou et al. proposed three carriers and the respective sample preparation methods are described for the determination of ethanol in human urine samples. The authors claim that the microscope slide with a cavity coated with gold substrate and covered with transparent cling film has succeeded a quick, simple and inexpensive identification of ethanol in the biological samples, offering a quite low limit of detection (LoD) of the volatile analyte. The paper is within the scope of Molecules.

The main conclusion seems to be supported by the data, while some details are unclear. I hope the following comments may help to improve the quality of the paper for future submission.

  1. This paper describes the comparison of the carriers designed by yourselves, whether it is compared with other methods for detecting ethanol, and whether it has advantages?
  2. The sample measured in this paper is urine with known alcohol concentration. Can this method be applied to real samples to determine the alcohol level in human urine after drinking alcoholic beverages (such as wine or beer)?
  3. Table 4 is incomplete and lacks the bottom border. It is recommended to separate the different factors with lines to make the table clearer.
  4. In P18 line 37, it is mentioned that the cleaning steps of the carriers are simple, whether the carrier has any influence on the sample test after cleaning?
  5. The conclusion and prospect section should discuss the influence and value of this method on future research.

Round 2

Reviewer 2 Report

The revised manuscript is significantly improved relative to the original.  The introduction provides better information on the target detection limits for this analysis, and a more reasonable comparison to other methods.  The limits of detection are now presented more quantitatively.  There remain some relatively minor concerns with the manuscript as described below.

The introduction now includes a section that indicates the relevant concentrations of ethanol in bodily fluids, as well as a more reasonable evaluation of the capabilities of headspace GC for the analysis of ethanol in blood.  The authors have also removed text stating that the proposed and evaluated vibrational spectroscopy methods could serve as a substitute for official methods.  All of this represents a more effective and reasoned comparison of the proposed methods to current state of the art.  However, it also leaves the reader curious as to the significance and utility of the proposed approach.  Blood alcohol analysis is typically employed in a forensic setting (it is likely the most frequent analysis run at forensic laboratories), where “official” methods capable of quantifying the ethanol level at and below the legal limit are indispensable.  The authors should clarify how they envision their approach being applied in a practical setting, perhaps as a nondestructive screening tool before more quantitative “official” approaches are applied.

Although the authors provide some indication of measurement to measurement reproducibility for their method in the response to reviewers, this is not included in the revised manuscript.  It should be.

The authors have indicated in the response to reviewers that the conclusions section has been significantly reduced in length by removing information already presented in the results section.  However, the conclusions section in the revised manuscript has not been reduced in length (if anything it is actually longer) and remains repetitive with the results section.  The conclusions should summarize the results as in Table 4, but should not present all of the details already presented in the results section.

In the conclusions it is important for the authors to distinguish between LOD, which is low enough to detect ethanol in urine at relevant levels, and limit of quantitation, which will be higher and well above relevant levels.  A method with an LOD similar to the relevant regulatory concentration could be of use as a screening tool, but would not be useful as a quantitative approach at that level with a reasonable level of uncertainty.

Reviewer 3 Report

The significance of this manuscript/study has not been enhanced by the amendments made by the authors, and the responses only address in part the issues raised during the initial review

Comment (i) - responses to this have at least put the measurement in the context of current practice

Comment (ii)  - original references to describe the range of literature on substrates for measuring liquids with Raman should be provided.

Comments (iii) & (iv) optimisation of the focal depth of the objective for each measurement geometry is the issue here, not optimisation of the laser focus for a single choice of objective.

Comment (v) I do not accept this - the water peak at ~1630 cm-1 should be visible as the dominant peak, and if the substrate is only partially reflective, then the study should include a dependence of performance on surface reflectivity.

Comment (vii)  - the LOD determined features strongly in the work, and therefore the most accurate determination IS within the scope of this work.

Comment (viii) The full spectrum should be shown for the cuvette method.

Comment (ix) I do not accept this, or at least it should be discussed.

Reviewer 4 Report

All questions were answered and the manuscript was modified following the reviewers' comments, thus I recommend publication of the manuscript. 

Author Response

No action required. Thank you